# Study on the Flammability, Crystal Behaviors and Mechanical Performance of Polyamide 11 Composites by Intercalated Layered Double Hydroxides

**DOI:** 10.3390/ijms232112818

**Published:** 2022-10-24

**Authors:** Cun Peng, Hua Yang, Wufei Tang

**Affiliations:** Hunan Engineering Technology Research Center for Comprehensive Development and Utilization of Biomass Resources, College of Chemistry and Bioengineering, Hunan University of Science and Engineering, Yongzhou 425199, China

**Keywords:** flammability, polyamide 11, layered double hydroxides, crystal behaviors

## Abstract

Sulfamic acid-intercalated MgAl-LDH (SA-LDH) was prepared by an anion exchange method, and its structure was characterized by X-ray diffraction (XRD) and Fourier transform infrared spectroscopy (FTIR). SA-LDH was introduced into polyamide 11 (PA11) by melt blending and to enhance the flame retardancy and mechanical properties. The scanning electron microscope (SEM) and XRD data showed that the lamellar structure of SA-LDH was partly disrupted. The cone calorimeter (CCT) results demonstrated that SA-LDH could effectively decrease the value of heat release rate, which may be ascribed to the better distribution of SA-LDH compared to LHD in the PA11 matrix. The effects of SA-LDH on the crystal behaviors of PA11 were investigated by XRD and differential scanning calorimetry (DSC), indicating that SA-LDH could induce the formation of new crystal forms and served as a heterogeneous nucleating agent. The mechanical progress caused by the incorporation of SA-LDH was correlated with compatibility improvement between SA-LDH and PA11.

## 1. Introduction

Polyamide 11 (PA11) is a biomass-stemmed polymer as its monomer, 11-amino- undecanoic acid is obtained from castor oil [1,2,3]. Similar to other PA materials, PA11 exhibits excellent mechanical properties, especially higher impact strength below −40 °C, superior abrasive resistance, and about 5~6 times-lower moisture absorption than PA6 or PA66. PA11 is widely used as an engineering and electrical or electronic equipment material [4,5,6].

On the other hand, considering the further application fields of PA materials, it is indeed necessary to improve the flame retardancy [7,8]. In the recent literature, nano-sized particles with lamellar structure have exhibited inspiring improvement to the melt drop and decreasing heat release during combustion [9,10].

Layered double hydroxides (LDH) have shown good flame retardancy and smoke suppression with nontoxic and halogen-free properties due to their unique chemical composition and layered structure [11,12,13,14,15,16,17,18]. For the special layered structure with exchangeable anion between layers, some researchers have attempted to intercalate flame-retardant elements to enhance its efficiency [19,20]. LDHs intercalated with flame retardant anions may combine the advantages of both Mg-Al metal salts and P/N flame retardants. During combustion, LDH loses the interlayer water to dilute the concentration of O_2_, and its intercalated anions can be dehydroxylated to become a mixed metal oxide. In addition, a huge amount of heat is adsorbed during combustion, protecting the bulk polymer from exposure to air and suppressing smoke production [21,22,23].

LDH have been used as flame retardants in PP [24,25,26] and EVA [27,28], but little efforts in PA11 have been reported. Some studies have documented that low-content LDHs (≤5%) can improve the mechanical performance of PA6 [29], PA66 [30], and PA11 [31]. It has been proposed that nano-sized particles can change the crystal behavior of PA and hence improve the mechanical performance [32,33].

SA and its derivates have been widely used as flame retardants and have also shown excellent flame retardancy in PA6 [34]. In this work, a novel sulfamic acid-intercalated LDH (SA-LDH) was prepared and introduced into PA11. The influence of LDH and SA-LDH to the structure and properties of PA11 are compared in detail in this article.

## 2. Results and Discussion

### 2.1. Characterization of the Structure of SA-LDH

The change of lamellar structure of LDH molecules after intercalation by SA can be reflected by FTIR spectra in Figure 1 and the XRD patterns in Figure 2.

The FTIR spectra of SA, LDH, and SA-LDH are shown in Figure 1. It can be seen that both LDH and SA-LDH showed the absorption bands of O-H stretching at about 3450 cm^−1^, δ(H-OH) vibrations at about 1625 cm^−1^, and the lattice vibration of the M-O and O-M-O (M=Mg and Al) groups in the region below 800 cm^−1^.

The peak at 1370 cm^−1^ in the spectrum of LDH was assigned to the carbonate stretching, which disappeared in the spectrum of SA-LDH with several newly emerged peaks. The peaks at 3320 and 3268 cm^−1^ were due to the stretching vibration of -NH_2_. The peaks at 1222, 1129, and 1066 cm^−1^, which corresponded to the vibrations of S=O in SA, also appeared in SA-LDH, which documented the existence of NH_2_SO_3_^−^ in SA-LDH.

Figure 2 shows the XRD patterns of LDH and SA-LDH. The pristine LDH presented a basal spacing of 0.76 nm (2*θ* = 11.5°), which was enlarged to 0.91 nm (2*θ* = 9.7°) after the intercalation by SA. The increase in basal spacing indicated that NH_2_SO_3_^-^ had been successfully intercalated into the interlayers of LDH. According to Equation (5), the intercalation ratio reached approximately 98%.

### 2.2. The Distribution of SA-LDH in PA11 Matrix

The distribution of either LDH or SA-LDH is reflected by the XRD patterns in Figure 3 and the SEM images show in Figure 4.

The special diffraction peaks of both LDH and SA-LDH remained in the composites, meaning the lamellar structure of either LDH or SA-LDH was not absolutely disrupted. According to the data tabulated in Table 1, the interplanar spacing (*d*) in LDH showed no change and the crystallite size *<D>* decreased bit, meaning the crystal structure of LDH changed little in PA11; however, a slight increase of *d* in SA-LDH indicated the lattice planes were enlarged slightly. This may have been caused by the entering of PA11 chains. Moreover, the apparent decrease of *<D>* documented the crystallization was partly disrupted with the abscission of some lattice planes. The number of deciduous lattice planes (*n*) was about 1~2 for LDH but 8~10 for SA-LDH in PA11, which resulted in the partial exfoliation of SA-LDH among the PA11 matrix.
*N* = (<*D*>_0_ − <*D*>*_p_*)/*d*(1)*<D>*_0_, crystallite size of sole LDH or SA-LDH; *<D>_p_*, crystallite size of LDH or SA-LDH in PA11; *D*, layer spacing of LDH or SA-LDH in PA11.

Another change that cannot be ignored was the diffraction peaks for PA11. As shown in Figure 3, only the reflection peak at 2*θ* ≈ 21.4° could be found in PA11, but in PA11/LDH, the reflection peaks appeared at 2*θ* ≈ 20° and 24°. In general, two main crystalline forms of α and γ in PA11 materials existed. The α form was a monoclinic unit cell with a zig-zag planar conformation with two reflection peaks at 2*θ* ≈ 20° and 24°; the γ form was characterized by a parallel arrangement of the chains with hydrogen bonds in which the amide groups are twisted out of the zig-zag planes, with the reflection peak at 2*θ* ≈ 21.4° [35]. The results mean γ form was inclined to form in PA11 by extrusion; however, the existence of either LDH or SA-LDH could induce the generation of the α form.

It was therefore suggested that the change of crystal form could ultimately affect the physical and mechanical properties. The α form exhibited a higher modulus below *T*_g_ but a more rapid decreased above *T*_g_ [36]. The γ form had a higher heat distortion temperature. Thus, the relative fraction of the crystalline phases in polymer will bring mechanical properties change accordingly.

The dispersion of LDH and SA-LDH was also observed by SEM. As shown in Figure 4, the scatter of LDH and SA-LDH in PA11 was obviously different. LDH molecules were inclined to aggregate with an approximate single particle size of 100~200 nm, which was much larger than the value calculated from XRD patterns in Table 1. The observed size of SA-LDH in PA11 from SEM was about 20~30 nm, which was consistent with the XRD calculation in Table 1. The SEM observation further illustrated a better distribution of SA-LDH than LDH in PA11.

### 2.3. Flame Retardancy

The flame retardancy of PA11 containing LDH or SA-LDH was compared with the neat PA11. Table 2 shows the LOI and UL-94 results, and Table 3 and Figure 5 show the cone results.

It was easy to find that an LDH concentration lower than 5% provided limited increment on the LOI, but equal SA-LDH contributed more than two increments. No improvement on the UL-94 grade was found for the two additives.

The result was similar to previous studies on the contribution of clays to the flame retardant. [37] The sole presence of scattered clays provided limited improvement to LOI and UL-94. Similarly, LDH provided little contribution on LOI, but SA was an effective flame retardant. Therefore, SA-LDH showed some enhancement on LOI.

Based on the HRR curves shown in Figure 5, one can see that after the addition of LDH or SA-LDH, the HRR curves became wider and shifted slightly right. The tendency to form two peaks can be found through the addition of 5 phr SA-LDH, reflecting that the time to pHRR was distinctly prolonged, from 161 s in PA11 to 237 s after 5 SA-LDH introductions. A slight decrease on the pHRR and slight increase on the THR at same time can be found from the relative data in Table 3.

According to previous studies [38,39], plate-like clays are inclined to form a barrier to prevent the release of heat and oxygen and often exhibit improvement during cone test (CCT). In this work, lamella-scattered LDH promoted the formation of protective charred layers to reduce the escaping ‘fuel.’, As a result, pHRR was reduced and combustion was spread out over time, which was also quantifiably described by a parameter in CCT, FGI (fire growth index), which was the ratio of pHRR to T_pHRR_ (pHRR/T_pHRR_) and reflects the spread rate of fire.

### 2.4. Crystal Behaviors

From the heating DSC traces shown in Figure 6, it can be observed the melt temperature of PA11 decreased slightly after the addition of LDH or SA-LDH. However, according to the data listed in Table 4, the maximal decrease was 1.3 °C, and the quantity of LDH or SA-LDH provided limited change on the melt temperature. Differently, the cooling curves in Figure 6 and data in Table 4 indicate that 2.5 or 5 phr LDH was 3.4 or 4.4 °C higher in crystal temperature compared with PA11. Similarly, 2.5 or 5 phr SA-LDH was 1.2 or 3.2 °C higher, implying that the crystallization happened earlier.

The earlier crystallization indicates that either LDH or SA-LDH can serve as a nucleation agent; on the other hand, the decrease on Δ*H_m_* and Δ*H_c_* may have also been caused by the restriction of rigid LDH or SA-LDH molecules to the movement of flexible polymer chains [40].

The isothermal crystallization kinetics of PA11 and PA11/SA-LDH nanocomposites were studied using DSC and described quantitatively according to the Avrami equation, written as:(2a)$$X_{t}=X_{c}(t)/X_{c}(\infty )=1-\exp(-Kt^{n})$$
where *X_t_* is the fraction of polymer crystallized at time ‘*t*’, and ‘*n*’ is a parameter that describes the nucleation and growth process. ‘*K*’ is the crystallization rate constant. However, the Avrami equation can be linearized as following form:(2b)$$\ln(-\ln(1-X_{t}))=n\ln t+\ln K$$

By linear regression, the values of “*n*” and “*K*” can be obtained according to the slope and intercept, respectively. Using the values of the kinetics parameters above, *t*_1/2_, the crystallization half time, defined as the time at which the extent of crystallization was 50%, can be obtained.
(3)$$t_{1/2}={\;(\ln2/K)}^{1/n}$$

According to the cooling curve of neat PA11 during the crystal process, four points at 167, 169, 171, and 173 °C were selected, respectively, as the appropriate isothermal crystal temperatures. The relative crystallinity and crystallization time of PA11 and its composites is shown in Figure 7 and the value calculated above is listed in Table 5.

For the neat PA11, the value of *n* increased from 1.2 to 3.1 with the increase of crystal temperature, implying that the crystal growth of PA11 did not absolutely follow three-dimensional spherulitic propagation. A similar trend existed in PA11 composites, with a lower *n* at same crystal temperature. This may be attributed to the fact that rigid particles serve as heterogeneous nucleating agent to change the nucleation mechanism. However, the changing trend of *n* was not similar for LDH and SA-LDH loading. For LDH, the decrease of *n* was consistent with the increase on LDH loading, but the opposite was observed in SA-LDH. The possible reason may be the different dispersion of LDH and SA-LDH in PA11 that was discussed above based on the XRD results, resulting in the aggregation of LDH but partial exfoliation of SA-LDH in PA11. In other words, SA-LDH induced denser nucleation of PA11; as a result, the crystal growth of PA11 was restrained and the dimension was also decreased.

The drastic drop of the crystallization rate parameter *k* and prolongation in crystallization half-time *t*_1/2_ with increased of crystal temperature indicated both nucleation and a slowed crystal growth rate of PA11 in higher temperature. This can be attributed to the more active mobility of polymer chains in higher temperature ranges, which was more difficult to fix.

The results document that LDH and SA-LDH played two competing roles during the crystallization process. On one hand, they served as heterogeneous nucleating agent to promote the crystallization, reflected by the earlier beginning crystallization, shorter of *t*_1/2_, and increased crystallization rate parameter *K* in the Avrami equation compared with the neat PA11. On the other hand, as rigid particles, both LDH and SA-LDH restrained the movement of polymer chains that retarded the crystal growth, reflected by the declined Δ*H_m_* and Δ*H_c_* values.

### 2.5. Mechanical Properties

The mechanical performance change brought by LDH or SA-LDH is reflected by the data in Table 6. It can be seen the incorporation of LDH or SA-LDH provided a slight improvement on the tensile stress but slightly decreased elongation at same time. The result is consistent with other polyamide-clay composites [26]. Both reinforcement and embrittlement were caused by rigid particles introduction.

For comparison of SA-LDH and LDH, equal SA-LDH addition brought higher tensile strength and elongation at break, possibly caused by the better compatibility between SA-LDH and PA11, which was reflected by SEM of smaller distribution size of SA-LDH in PA11.

## 3. Materials and Methods

### 3.1. Materials

Polyamide11 (LOT142170-75) was provided by Arkema, Paris, France. MgAl-CO_3_^2^ -LDH (LDH) was offered from Nan Tong Adwance Chemicals Ltd., Nantong, China. Sulfamic acid (SA) was offered from Xi Long Chemicals Ltd., Guangzhou, China.

### 3.2. Preparation of SA-LDH

First, 10 g LDH was dispersed in 200 mL deionized water with vigorous stirring in a 500 mL three-necked distilling flask, and then the SA solution (10 g SA/50 mL deionized water) was added into the slurry above and the mixture was stirred vigorous at 50 °C for 2 h by adding drop by drop (nearly none bubbles could be founded). Lastly, the precipitate was finally washed, filtered, and dried at 50 °C for 10 h. The product was referred to sulfamic acid-intercalated LDH (SA-LDH).

### 3.3. Sample Preparation

The PA11 composites was melt-compounded on a Thermo Scientific HAAKE Rheomix OS PTW, including a co-rotating two-screw extruder (barrel length 400 mm and screw diameter L/D = 25) equipped with a feed-dosing element. The temperature setting of the 10 zones in extruder from the hopper to die was 230/240/245 °C, and the screw speed was 35 rpm. The extrudate was first pelletized and then ground to powder in liquid nitrogen in an ultracentrifuge mill. At last, the powder was dried for 12 h under 80 °C before being injected to the suitable size for limited oxygen index (LOI) and UL-94 vertical burning tests by special injection molders.

### 3.4. Characterizations

X-ray diffraction: XRD patterns were recorded by a D/max2500 VB2+/PC X-ray diffractometer. The Cu *K_α_* radiation source (*λ* = 0.154 nm) was operated at 40 KV and 20 mA. Patterns were recorded by monitoring those diffractions appearing from 3~80°. The scan speed was 5°/min. The average layer spacing was calculated by Bragg Equation (4):2*dsinθ* = *nλ*(4)
where *d* is the basal spacing, *θ* is the diffraction angle, and *n* = 1.

The intercalation ratio was calculated by Equation (5):(5)$$\langle IR\rangle =A_{003}/\left(A_{003}+{A’}_{003}\right)$$
where *A*_003_ and *A′*_003_ refer to the diffraction peak area of (003) crystal plane in the XRD patterns of SA-LDH after and before intercalation, respectively. In other words, *A*_003_ and *A′*_003_ mean the content of SA-LDH and residual LDH, respectively.

The crystallite size can be obtained using the Derby-Scherrer Equation (6):(6)$$\;\langle D\rangle =k\lambda /\left(\beta \cos\theta \right)$$
where *D* is the crystallite size in the normal direction of the selected set of crystal planes, *2θ* is the Bragg reflection angle of the corresponding set of crystal planes, *k* is the Derby-Scherrer constant (*k* = 0.9), *λ* is the X-ray wavelength (*λ* = 0.154 nm for Cu-K*α* radiation), *β* is the calibrated half-width of Bragg reflection peak (*β* = (*B*^2^ − *b_0_*^2^)^1/2^, *B* is the experimental half-width of the peak, and *b*_0_ is the instrumental broadening factor, hereby *b_0_* = 0.16°. Apparently, *D* is inversely proportional to *β* when *θ* is given, and vice versa.

FT-IR spectra were collected from a Nicolet IS5 FTIR under the resolution of 1 cm^−1^ in 128 scans, and the correlative powders were studied using KBr pellets.

The sample dimensions for LOI and UL-94 tests were 100 × 6.5 × 3 and 125 × 12.5 × 3 mm^3^, according to ISO standard 1210, 1992 and ASTM D3801-1996, respectively.

A fire testing technology cone calorimeter was also used to evaluate the fire performance of the composites according to the standard ISO 5660 under a heat flux of 50 kW/m^2^ with a size of 100 × 100 × 3 mm^3^, which was comparable to that of a mild fire scenario. Specimens were wrapped in aluminum foil, leaving the upper surface exposed to the radiator, and then placed on ceramic backing board at a distance of 25 mm from cone base. The experiments were repeated five times and a series of parameters were obtained, including the time required to ignition, heat release rate (HRR), peak heat release rate (pHRR), and total heat released (THR).

SEM observation was conducted by means of a field-emission scanning electron microscope Hitachi S-4700 under the voltage of 20 kV. With each sample fractured in liquid nitrogen and covered with Pt, more than 10 SEM images were taken at different magnifications to assess the true morphology.

The crystallization behavior was described by a DSC TA Q100. Each sample (about 3~5 mg) was first heated to 200 °C at 10 °C/min, and then held at 200 °C for 5 min to eliminate the thermal history. The cooling curve was recorded from 200 to 40 °C at a rate 10 °C/min. Eventually, the second heating curves was proceeded at 10 °C/min from 40 to 200 °C, exhibiting the trace of melting process.

The effect of incorporating LDH and SA-LDH on the crystal behavior of PA11 was further investigated by DSC. Isothermal crystallization was carried out by DSC curves. Samples were heated from 40 to 200 °C at 10 °C/min; held at 200 °C for 5 min to ensure complete melting; then cooled to 167, 169, 171, and 173 °C at a rapid rate of 80 °C/min; and held until the whole crystallization process was complete. The selection of crystal temperature referred to the crystal temperature scope reflected by the cooling curves.

The mechanical tests were carried out on a UMT-1422 tensile testing machine (Jinjian instrument Co., Chengde, China) according to the ISO 527-1 standard. The samples were tested with a speed of 50 mm/min. Five parallel samples for each specimen were tested to obtain an averaged value.

## 4. Conclusions

SA-LDH was prepared by anion exchange. New groups of -NH_2_ and S=O emerged in the FTIR spectra, and interplanar spacing enlargement reflected by XRD demonstrated that SA was successfully intercalated into LDH interlayers. Calculations according to XRD and SEM observation both evidenced the better dispersion of SA-LDH in PA11 than LDH. The contribution of SA-LDH on flame retardant was mainly reflected by the prolongation of heat release during combustion. The change of crystal behaviors, including the crystal form and crystallization kinetics brought by SA-LDH or LDH to PA11, were correlated with mechanical performance improvement.

## Figures and Tables

**Figure 1 ijms-23-12818-f001:**
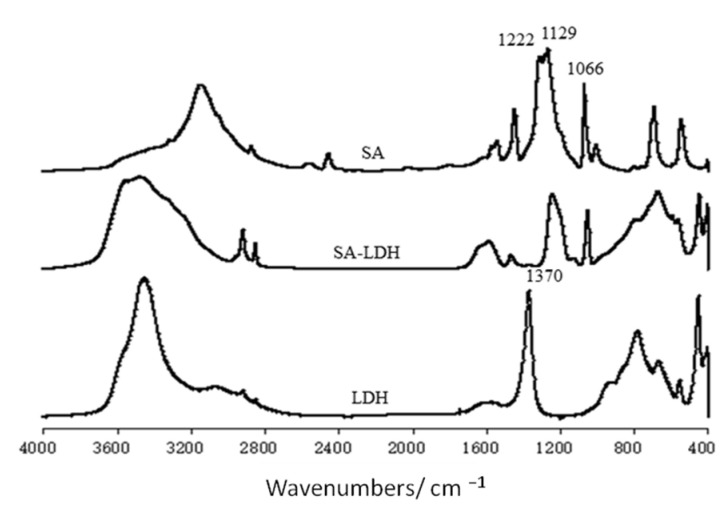
FT-IR spectra of LDH, SA-LDH, and NH_2_SO_3_H.

**Figure 2 ijms-23-12818-f002:**
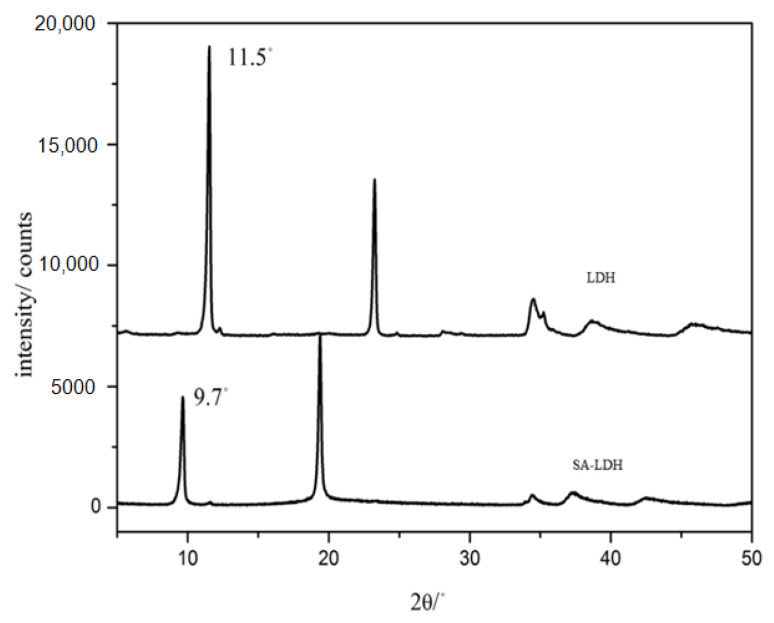
XRD patterns of LDH and SA-LDH.

**Figure 3 ijms-23-12818-f003:**
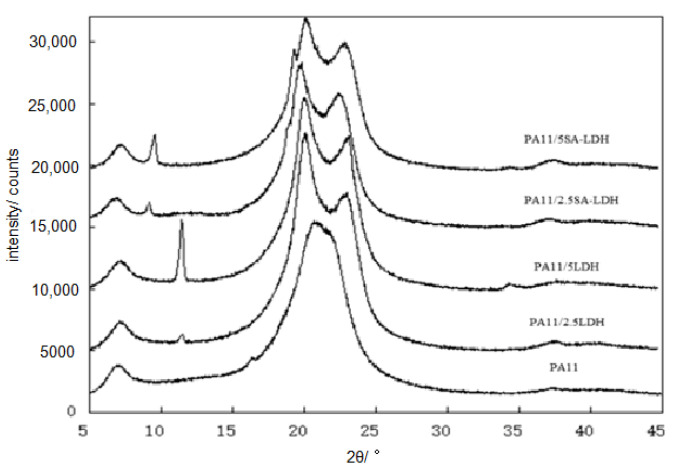
XRD patterns of PA11 and its composites.

**Figure 4 ijms-23-12818-f004:**
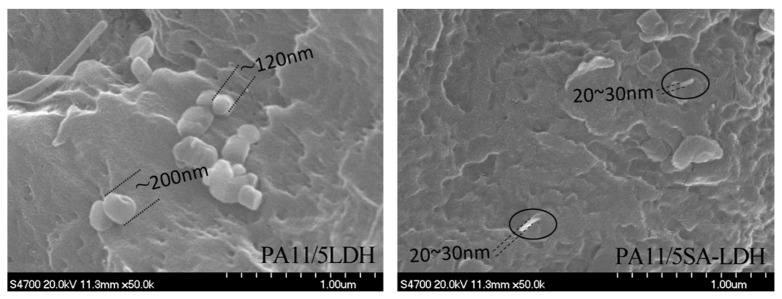
SEM images of PA11/LDH and PA11/SA-LDH composites.

**Figure 5 ijms-23-12818-f005:**
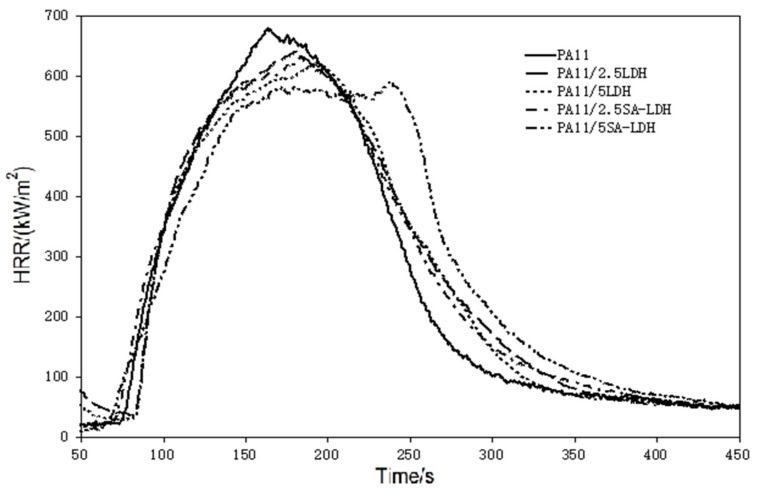
HRR curves of PA11 and its composites.

**Figure 6 ijms-23-12818-f006:**
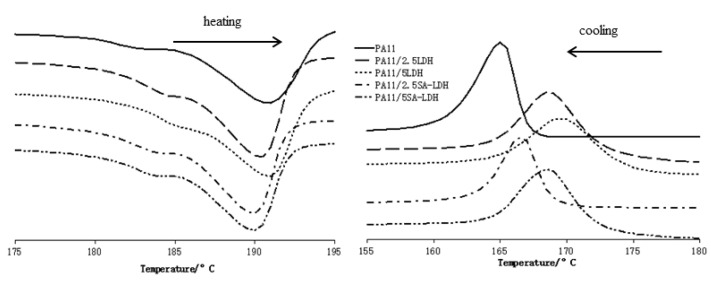
DSC curves of PA11 and its composites.

**Figure 7 ijms-23-12818-f007:**
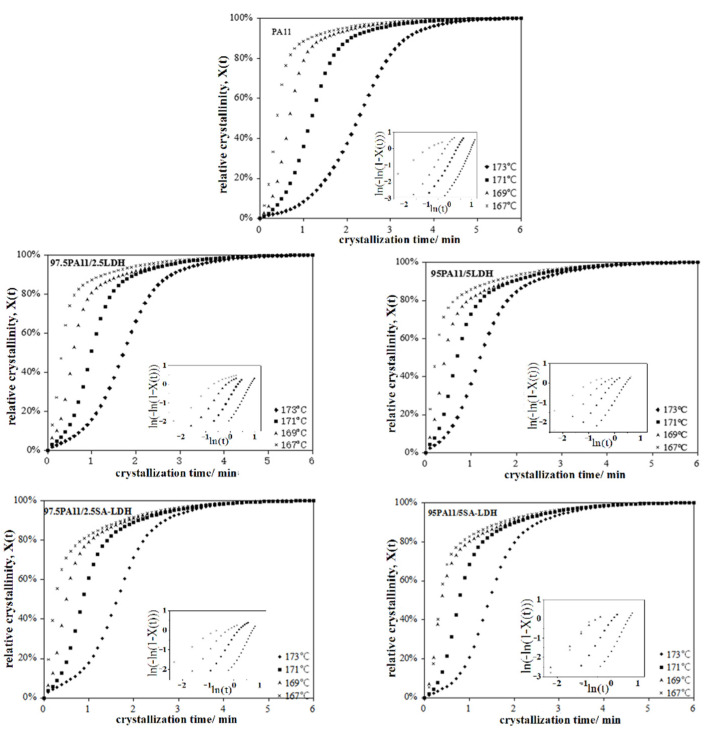
Relationship between the relative crystallinity and crystallization time of PA11 and its composites.

**Table 1 ijms-23-12818-t001:** XRD data of PA11 and its composites.

	LDH (003)			SA-LDH (003)		
	LDH	PA11/2.5LDH	PA11/5LDH	SA-LDH	PA11/2.5SA-LDH	PA11/5SA-LDH
*2θ/°*	11.5	11.5	11.5	9.7	9.2	9.6
*d*/nm	0.77	0.77	0.77	0.91	0.96	0.92
*<D>*/nm	47.7	46.3	47.0	36.9	29.3	27.8
*n **		1.8	0.9		7.9	9.9

** n*, number of deciduous lattice planes.

**Table 2 ijms-23-12818-t002:** LOI and UL-94 results for PA11 and its composites.

	LOI (vol%)	UL-94
PA11	24.9 ± 0.2	No rating
PA11/2.5LDH	25.0 ± 0.1 (↑0.1)	No rating
PA11/5LDH	25.3 ± 0.2 (↑0.3)	No rating
PA11/2.5SA-LDH	27.2 ± 0.3 (↑2.3)	No rating
PA11/5SA-LDH	27.5 ± 0.2 (↑2.5)	No rating

**Table 3 ijms-23-12818-t003:** Some key results for PA11 composites of CCT.

	pHRR/(kW/m^2^)	TTP (T_PHRR_)/s	THR/(MJ/m^2^)	FGI*/(kW/m^2^•s)
PA11	678.7 ± 17	161 ± 3	93.2 ± 15	4.2
PA11/2.5LDH	641.1 ± 15 (↓5.5%)	180 ± 4 (↑19s)	101.7 ± 16 (↑9.1%)	3.6
PA11/5LDH	617.3 ± 12 (↓9.0%)	191 ± 3 (↑30s)	99.2 ± 12 (↑6.4%)	3.2
PA11/2.5SA-LDH	628.7 ± 12 (↓7.3%)	184 ± 3 (↑23s)	101.2 ± 11 (↑9.0%)	3.4
PA11/5SA-LDH	592.9 ± 11 (↓12.6%)	237 ± 3 (↑76s)	101.8 ± 10 (↑9.1%)	2.5

* FGI = pHRR/T_pHRR_; TTP = Time to pHRR.

**Table 4 ijms-23-12818-t004:** DSC data for PA11 and its composites.

	*T_m_*/°C	*T_c_*/°C	(*T_m_* − *T_c_*)/°C	Δ*H_m_*/(J/g)	Δ*H_c_*/(J/g)
PA11	190.6	165.3	25.3	41.3	44.4
PA11/2.5LDH	190.4	168.7	21.7	40.1	37.6
PA11/5LDH	190.9	169.7	21.2	40.0	33.8
PA11/2.5SA-LDH	189.7	166.5	23.2	40.8	40.3
PA11/5SA-LDH	189.9	168.7	21.2	38.5	37.2

**Table 5 ijms-23-12818-t005:** Some parameters from the Avrami equation for PA11 and its composites.

		*T_c_*/°C			
		167	169	171	173
*n*	PA11	1.2	2.2	2.5	3.1
	PA11/2.5LDH	1.2	2.0	2.4	2.7
	PA11/5LDH	1.2	1.4	1.9	2.2
	PA11/2.5SA-LDH	1.2	1.5	2.0	2.8
	PA11/5SA-LDH	1.2	1.6	2.3	2.9
*K*	PA11	3.4	1.6	0.5	0.06
	PA11/2.5LDH	3.4	2.0	0.7	0.2
	PA11/5LDH	4.1	2.2	1.3	0.5
	PA11/2.5SA-LDH	3.4	1.8	0.9	0.2
	PA11/5SA-LDH	5.5	2.9	1.2	0.2
*t* _1/2_	PA11	0.3	0.7	1.2	2.2
	PA11/2.5LDH	0.3	0.6	1.0	1.7
	PA11/5LDH	0.2	0.4	0.7	1.2
	PA11/2.5SA-LDH	0.3	0.5	0.9	1.6
	PA11/5SA-LDH	0.3	0.4	0.8	1.4

**Table 6 ijms-23-12818-t006:** Summary of mechanical properties for PA11 and its composites.

Samples	Tensile Strength/Mpa	Elongation at Break/%
PA11	41.6 ± 3	264 ± 6
PA11/2.5LDH	42.3 ± 3	222 ± 4
PA11/5LDH	42.1 ± 1	197 ± 4
PA11/2.5SA-LDH	46.3 ± 2	230 ± 3
PA11/5SA-LDH	47.1 ± 2	205 ± 4

## Data Availability

The data are available upon request.

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
