# Peer review of "Study on the Flammability, Crystal Behaviors and Mechanical Performance of Polyamide 11 Composites by Intercalated Layered Double Hydroxides"

_ijms, 2022, doi:10.3390/ijms232112818_

Round 1

Reviewer 1 Report

Authors reported “Study on the flammability, crystal behaviors and mechanical performance of polyamide 11 composites by intercalated layered double hydroxides”. The discussion is well organized. The characterization and calculation are both solid for the conclusion. However, there are problems with the experimental design. I have some comments which are listed below:

1 The originality of a dissertation is insufficient.

2 How acid intercalation is used to overcome the reaction of acid with LDH and surface modification of LDH?

3 The authors need to revise some obvious errors should be carefully checked and corrected in the text. For example: Figure 4 and Table 6.

4 Lack of standard for mechanical properties of samples.

Author Response

Response to reviewer

We would like to thank the reviewer for his comments, and we have carefully revised the manuscript accordingly. All changes have been marked in the revised manuscript.

Authors reported “Study on the flammability, crystal behaviors and mechanical performance of polyamide 11 composites by intercalated layered double hydroxides”. The discussion is well organized. The characterization and calculation are both solid for the conclusion. However, there are problems with the experimental design. I have some comments which are listed below:

1, The originality of a dissertation is insufficient.

Response: Thanks for your comments.

In this work, sulfamic acid (SA) was selected to obtain the intercalated LDH (SA-LDH). The structure SA-LDH were analyzed, and then introduced into PA11 to mainly study on the flammability, crystal behaviors and mechanical properties. In reported work, researchers paid more attention to use other organic materials intercalated LDH to improve the flame retardancy and other performances (Chemosphere 2021, 277, 130370; Journal of Materials Science 2017, 52: 12235-12250; Applied Clay Science 2020, 187, 105481.); Our team has reported SA intercalated into the layers of LDH, but the application field was ethylene vinyl acetate (EVA), PLA/intumescent flame retardant and PA11/aluminum phosphinate composites (Materials Letters 2015, 150, 31-34; Fibers and Polymers 2017, 18, 2060-2069; Journal of Applied Polymer Science 2016, 132, 41761.), respectively.

Anyway, although polymer composites containing intercalated LDH are an old and ongoing research topic, there is a limited knowledge available on SA intercalated LDH, as well as what role it plays in flame-retardant, crystal behaviors and mechanical properties of PA11 composites.

The relevant references are as following:

1, Liu, Y.; Gao, Y.; Zhang, Z.; et al. Preparation of ammonium polyphosphate and dye co-intercalated LDH/polypropylene composites with enhanced flame retardant and UV resistance properties. Chemosphere 2021, 277, 130370.

2, Jin, X.; Gu, X.; Chen, C.; et al. The fire performance of polylactic acid containing a novel intumescent flame retardant and intercalated layered double hydroxides. Journal of Materials Science 2017, 52: 12235-12250.

3, Jin, L.; Zeng, H.; Du, J.; et al. Intercalation of organic and inorganic anions into layered double hydroxides for polymer flame retardancy. Applied Clay Science 2020, 187, 105481.

4, Jiang, Y.; Gu, X.; Zhang, S.; et al. The preparation and characterization of sulfamic acid-intercalated layered double hydroxide. Materials Letters 2015, 150, 31-34.

5, Zhang, S.; Tang, W.; Gu, X.; et al. Preparation and characterization of intumescent flame retardant biodegradable poly(lactic acid) nanocomposites based on sulfamic acid intercalated layered double hydroxides. Fibers and Polymers 2017, 18, 2060-2069.

6, Zhang, S.; Tang, W.; Gu, X.; et al. Flame retardancy and thermal and mechanical performance of intercalated, layered double hydroxide composites of polyamide 11, aluminum phosphinate, and sulfamic acid. Journal of Applied Polymer Science 2016, 132, 41761.

2, How acid intercalation is used to overcome the reaction of acid with LDH and surface modification of LDH?

Response: Thanks for your comments, and this is a professional question.

In this work, MgAl-CO32 -LDH (LDH) could easily react with SA by CO32 to produce CO2. So, it could observe the velocity and number of bubbles to judge whether the reaction was completed. Meanwhile, the corresponding contents modification were also revised by red mark in section “3.2. Preparation of SA-LDH” revise manuscript.

Modification:

Line 14 of Page 14: …at 50 oC for 2 h (nearly none bubbles could be founded). Lastly the precipitate was…

3, The authors need to revise some obvious errors should be carefully checked and corrected in the text. For example: Figure 4 and Table 6.

Response: Thank you for your good suggestions. We have carefully revised the errors.

Modification:

Figure 4. SEM images of PA11/SA-LDH and PA11/SA-LDH composites

Table 6. Summary of mechanical properties for PA11 and its composites

Samples

Tensile Strength/MPa

Elongation at break/%

PA11

41.6 ± 2.6

264 ± 5.5

PA11/2.5LDH

42.3 ± 3.4

222 ± 4.2

PA11/5LDH

42.1 ± 0.9

197 ± 4.4

PA11/2.5SA-LDH

46.3 ± 1.7

230 ± 3.4

PA11/5SA-LDH

47.1 ± 2.3

205 ± 4.4

4, Lack of standard for mechanical properties of samples.

Response: Thanks for your comments. The characterizations of mechanical properties of samples had added into the section “3.4 Characterizations”.

Modification:

The mechanical tests were carried out on a UMT-1422 tensile testing machine (Jinjian instrument Co., Chengde, China) according to ISO 527-1 standard. The samples were tested with a speed of 50 mm/min. Five parallel samples for each specimen were tested to obtain averaged value.

Reviewer 2 Report

This work shows some new insights, but there are still a few problems to be solved before publication. Please see the detailed comments below. After a minor revision, I would like to recommend this manuscript be published.

1. Line 12 on page 1 of section “Abstract”, I think the work “and” should be deleted, it's redundant.

2. A part of tense problems need to be corrected, such as line 3 on page 2, “was” should be “is”. Authors should carefully check again for this problem throughout the whole article.

3. Line 2 on page 6, the number of formulations is 4. Is it not 1? Please check again carefully in the manuscript. Meanwhile, the same problem should be corrected corresponding.

4. In Figure 4, the size data should be marked to facilitate reading for readers.

5. Lines 15-16 on page 6 and 15-16 on page 7 should be moved to line 6 on page 6 before.

6. Table 2, the LOI values should be added “vol%” as a unit.

7. 3.3, the information on LOI and UL-94 vertical burning tests should be moved to the 3.4 part.

8. The introduction part shows a little simpleness, especially for LDHs. The authors can cite some relevant references and add appropriate descriptions. For example, Chem. Commun. 2015, 51, 3024–3036.; Polym. Degrad. Stab. 2022, 204, 110104.; Crystals. 2020 Jul 14;10(7):612.; Chem. Rev. 2012, 112, 4124–4155.; Int. J. Mol. Sci. 2022, 23(19), 11049. 

Author Response

Response to reviewer

We would like to thank the reviewer for his comments, and we have carefully revised the manuscript accordingly. All changes have been marked in the revised manuscript.

This work shows some new insights, but there are still a few problems to be solved before publication. Please see the detailed comments below. After a minor revision, I would like to recommend this manuscript be published.

  1. Line 12 on page 1 of section “Abstract”, I think the work “and” should be deleted, it's redundant.

Response: Thanks for your carefully check. We had corrected this error in revise manuscript.

  1. A part of tense problems need to be corrected, such as line 3 on page 2, “was” should be “is”. Authors should carefully check again for this problem throughout the whole article.

Response: Thanks for your comments. We had corrected this error in revise manuscript.

  1. Line 2 on page 6, the number of formulations is 4. Is it not 1? Please check again carefully in the manuscript. Meanwhile, the same problem should be corrected corresponding.

Response: Thank you for your careful observation. We had carefully corrected those errors in revise manuscript.

  1. In Figure 4, the size data should be marked to facilitate reading for readers.

Response: Thanks for your comments. We have added the size data in Figure 4.

Modification:

Figure 4. SEM images of PA11/SA-LDH and PA11/SA-LDH composites

  1. Lines 15-16 on page 6 and 15-16 on page 7 should be moved to line 6 on page 6 before.

Response: Thank you for your good suggestions. We have made corresponding modifications.

  1. Table 2, the LOI values should be added “vol%” as a unit.

Response: Thanks for your comments. We have added the unit into Table 2.

Modification:

Table 2. LOI and UL-94 results for PA11 and its composites

LOI (vol%)

UL-94

PA11

24.9±0.2

No rating

PA11/2.5LDH

25.0±0.1 (↑0.1)

No rating

PA11/5LDH

25.3±0.2 (↑0.3)

No rating

PA11/2.5SA-LDH

27.2±0.3 (↑2.3)

No rating

PA11/5SA-LDH

27.5±0.2 (↑2.5)

No rating

  1. 3.3, the information on LOI and UL-94 vertical burning tests should be moved to the 3.4 part.

Response: Thank you for your good suggestions. We have moved the contents of information on LOI and UL-94 vertical burning tests to 3.4 part in revise manuscript.

  1. The introduction part shows a little simpleness, especially for LDHs. The authors can cite some relevant references and add appropriate descriptions. For example, Chem. Commun. 2015, 51, 3024–3036.; Polym. Degrad. Stab. 2022, 204, 110104.; Crystals. 2020 Jul 14;10(7):612.; Chem. Rev. 2012, 112, 4124–4155.; Int. J. Mol. Sci. 2022, 23(19), 11049. 

Response: Thanks for your comments. We have added the references in revise manuscript.

Modification:

  1. Zhou, L.; Li, W.; Zhao, H.; et al. NiTi-layered double hydroxide nanosheets toward high-efficiency flame retardancy and smoke suppression for silicone foam. Polymer Degradation Stability 2022, 204, 110104.
  2. Zhou, L.; Li, W.; Zhao, H.; et al. Comparative study of the M(â…¡)Al (M=Co, Ni) layered double hydroxides for silicone foam: characterization, flame retardancy, and smoke suppression. International Journal of Molecular Sciences 2022, 23, 11049.
  3. Mochane, M.; Magagula, S.; Sefadi, J.; et al. Morphology, thermal stability, and flammability properties of polymer-layered double hydroxide (LDH) nanocomposites: a review. Crystals 2020, 14, 612.
  4. Gu, Z.; Atherton, J.; Xu, Z. Hierarchical layered double hydroxide nanocomposites: structure, synthesis and applications. Chemical communications 2015, 51, 3024-3036.
  5. Wang, Q.; O’Hare, D. Recent advances in the synthesis and application of layered double hydroxide (LDH) nanosheets. Chemical Reviews 2012, 112, 4124-4155.

Round 2

Reviewer 1 Report

Authors reported “Study on the flammability, crystal behaviors and mechanical performance of polyamide 11 composites by intercalated layered double hydroxides”. The discussion is well organized. The characterization and calculation are both solid for the conclusion. However, there are problems with the experimental design.

I have some comments which are listed below:

1 The originality of a dissertation is insufficient.

2 How acid intercalation is used to overcome the reaction of acid with LDH and surface modification of LDH?

3 The authors need to revise some obvious errors should be carefully checked and corrected in the text. For example: Figure 4 and Table 6.

4 Lack of standard for mechanical properties of samples.

Author Response

Response to reviewer

We would like to thank the reviewer for his comments, and we have carefully revised the manuscript accordingly. All changes have been marked in the revised manuscript.

Authors reported “Study on the flammability, crystal behaviors and mechanical performance of polyamide 11 composites by intercalated layered double hydroxides”. The discussion is well organized. The characterization and calculation are both solid for the conclusion. However, there are problems with the experimental design. I have some comments which are listed below:

1, The originality of a dissertation is insufficient.

Response: Thanks for your comments.

In this work, sulfamic acid (SA) was selected to obtain the intercalated LDH (SA-LDH). The structure SA-LDH were analyzed, and then introduced into PA11 to mainly study on the flammability, crystal behaviors and mechanical properties. In reported work, researchers paid more attention to use other organic materials intercalated LDH to improve the flame retardancy and other performances (Chemosphere 2021, 277, 130370; Journal of Materials Science 2017, 52: 12235-12250; Applied Clay Science 2020, 187, 105481.); Our team has reported SA intercalated into the layers of LDH, but the application field was ethylene vinyl acetate (EVA), PLA/intumescent flame retardant and PA11/aluminum phosphinate composites (Materials Letters 2015, 150, 31-34; Fibers and Polymers 2017, 18, 2060-2069; Journal of Applied Polymer Science 2016, 132, 41761.), respectively.

Anyway, although polymer composites containing intercalated LDH are an old and ongoing research topic, there is a limited knowledge available on SA intercalated LDH, as well as what role it plays in flame-retardant, crystal behaviors and mechanical properties of PA11 composites.

The relevant references are as following:

1, Liu, Y.; Gao, Y.; Zhang, Z.; et al. Preparation of ammonium polyphosphate and dye co-intercalated LDH/polypropylene composites with enhanced flame retardant and UV resistance properties. Chemosphere 2021, 277, 130370.

2, Jin, X.; Gu, X.; Chen, C.; et al. The fire performance of polylactic acid containing a novel intumescent flame retardant and intercalated layered double hydroxides. Journal of Materials Science 2017, 52: 12235-12250.

3, Jin, L.; Zeng, H.; Du, J.; et al. Intercalation of organic and inorganic anions into layered double hydroxides for polymer flame retardancy. Applied Clay Science 2020, 187, 105481.

4, Jiang, Y.; Gu, X.; Zhang, S.; et al. The preparation and characterization of sulfamic acid-intercalated layered double hydroxide. Materials Letters 2015, 150, 31-34.

5, Zhang, S.; Tang, W.; Gu, X.; et al. Preparation and characterization of intumescent flame retardant biodegradable poly(lactic acid) nanocomposites based on sulfamic acid intercalated layered double hydroxides. Fibers and Polymers 2017, 18, 2060-2069.

6, Zhang, S.; Tang, W.; Gu, X.; et al. Flame retardancy and thermal and mechanical performance of intercalated, layered double hydroxide composites of polyamide 11, aluminum phosphinate, and sulfamic acid. Journal of Applied Polymer Science 2016, 132, 41761.

2, How acid intercalation is used to overcome the reaction of acid with LDH and surface modification of LDH?

Response: Thanks for your comments, and this is a professional question.

In this work, MgAl-CO32 -LDH (LDH) could easily react with SA by CO32 to produce CO2. So, it could observe the velocity and number of bubbles to judge whether the reaction was completed. Meanwhile, the corresponding contents modification were also revised by red mark in section “3.2. Preparation of SA-LDH” revise manuscript.

Modification:

Line 14 of Page 14: …at 50 oC for 2 h (nearly none bubbles could be founded). Lastly the precipitate was…

3, The authors need to revise some obvious errors should be carefully checked and corrected in the text. For example: Figure 4 and Table 6.

Response: Thank you for your good suggestions. We have carefully revised the errors.

Modification:

Figure 4. SEM images of PA11/ LDH and PA11/SA-LDH composites

Table 6. Summary of mechanical properties for PA11 and its composites

Samples

Tensile Strength/MPa

Elongation at break/%

PA11

41.6 ± 3

264 ± 6

PA11/2.5LDH

42.3 ± 3

222 ± 4

PA11/5LDH

42.1 ± 1

197 ± 4

PA11/2.5SA-LDH

46.3 ± 2

230 ± 3

PA11/5SA-LDH

47.1 ± 2

205 ± 4

4, Lack of standard for mechanical properties of samples.

Response: Thanks for your comments. The characterizations of mechanical properties of samples had added into the section “3.4 Characterizations”.

Modification:

The mechanical tests were carried out on a UMT-1422 tensile testing machine (Jinjian instrument Co., Chengde, China) according to ISO 527-1 standard. The samples were tested with a speed of 50 mm/min. Five parallel samples for each specimen were tested to obtain averaged value.
